# Transcriptome Analysis and Identification of Genes Associated with Cotton Seed Size

**DOI:** 10.3390/ijms25189812

**Published:** 2024-09-11

**Authors:** Bing Jia, Pan Feng, Jikun Song, Caoyi Zhou, Yajie Wang, Bingbing Zhang, Man Wu, Jinfa Zhang, Quanjia Chen, Jiwen Yu

**Affiliations:** 1Engineering Research Centre of Cotton, Ministry of Education, College of Agriculture, Xinjiang Agricultural University, 311 Nongda East Road, Urumqi 830052, China; jiabing1814@126.com (B.J.); fp2012622@163.com (P.F.); z459786119@126.com (C.Z.); 2State Key Laboratory of Cotton Biology, Cotton Institute of the Chinese Academy of Agricultural Sciences, Key Laboratory of Cotton Genetic Improvement, Ministry of Agriculture, Anyang 455000, China; sjkow513@163.com (J.S.); wangyj4624@163.com (Y.W.); 18439247312@163.com (B.Z.); wuman2004@163.com (M.W.); 3College of Agriculture, Tarim University, Alaer 843300, China; 4Department of Plant and Environmental Sciences, New Mexico State University, Las Cruces, NM 880033, USA; jinzhang@nmsu.edu

**Keywords:** upland cotton, seed size, RNA-seq, gene regulatory modules, *GhUXS5*

## Abstract

Cotton seeds, as the main by-product of cotton, are not only an important raw material for edible oil and feed but also a source of biofuel. The quality of cotton seeds directly affects cotton planting and is closely related to the yield and fiber quality. However, the molecular mechanism governing cotton seed size remains largely unexplored. This study investigates the regulatory mechanisms of cotton seed size by focusing on two cotton genotypes, N10 and N12, which exhibit notable phenotypic variations across multiple environments. Developing seeds were sampled at various stages (5, 20, 30, and 35 DPA) and subjected to RNA-seq. Temporal pattern clustering and WGCNA on differentially expressed genes identified 413 candidate genes, including these related to sugar metabolism that were significantly enriched in transcriptional regulation. A genetic transformation experiment indicated that the overexpression of the *GhUXS5* gene encoding UDP-glucuronate decarboxylase 5 significantly increased seed size, suggesting an important role of *GhUXS5* in regulating cotton seed size. This discovery provides crucial insights into the molecular mechanisms controlling cotton seed size, helping to unravel the complex regulatory network and offering new strategies and targets for cotton breeding to enhance the economic value of cotton seeds and overall cotton yield.

## 1. Introduction

Cotton, recognized as one of the pivotal agricultural commodities globally, is distinguished by the cellulose-rich fibers produced by its seed coat cells, which serve as the primary natural fiber utilized in the textile industry [1]. Concurrently, the cotyledons of cottonseeds are abundant in oils and proteins, rendering them not only suitable for the extraction of cottonseed oil but also for use as a raw material for producing protein-rich livestock feed [2]. Consequently, both cotton fibers and seeds are indispensable resources for societal development. Presently, cotton research predominantly focuses on enhancing fiber yield and quality, with comparatively less emphasis on seed quality. Traditional cotton breeding research has primarily aimed at increasing lint yield, which has inadvertently led to a trend of diminishing seed size [3]. Smaller seeds, due to their inadequate nutritional content, exhibit reduced viability, thereby constraining further improvements in fiber quality [4]. In the pursuit of cultivating high-quality cotton varieties, superior cotton seeds emerge as a critical factor that cannot be overlooked [5]. However, the molecular mechanisms and regulatory networks governing cotton seed size remain inadequately explored, thereby limiting the understanding and application of cotton seed quality improvements.

Seeds, serving as the reproductive entities of most plants, have long been acknowledged as a critically adaptive characteristic in terms of their size [6]. In angiosperms, the developmental trajectory of seeds commences with the process of double fertilization, which subsequently gives rise to a diploid embryo and a triploid endosperm [7]. The embryo and endosperm are encased within the seed coat, with the coordinated growth and regulation of these components ultimately determining the seed’s size [8]. The developmental process of seeds is also a manifestation of genetic influence, as evidenced by numerous genes that have been validated in the model crop *Arabidopsis* to regulate seed organ development [9,10]. Examples include genes associated with endosperm size, such as *HAIKU1* (*IKU1*) and *IKU2* [11,12,13]; genes influencing ovule integument cell growth, like *TRANSPARENT TESTA GLABRA2* (*TTG2*) [14] and *APETALA2* (*AP2*) [15]; and genes affecting embryo development, such as the simultaneous knockout of *ARABIDOPSIS HISTIDINE KINASE2* (*AHK2*), *AHK3*, and *AHK4*, which leads to the formation of large embryos within seeds [16]. Previous research has elucidated several regulatory mechanisms involved in seed development, including the IKU (HAIKU) pathway, the ubiquitin-proteasome pathway, G protein signaling pathways (guanosine triphosphate), mitogen-activated protein kinase (MAPK) signaling pathways, plant hormones, and transcriptional regulators [17]. These pathways and factors interact with one another, collectively orchestrating the developmental processes of seeds, and thereby impacting seed size and quality.

During the developmental process of cotton seeds, the concomitant growth and development of fibers and ovules are distinctive features that markedly differentiate cotton seed development from that of other seeds. The outer integument of cotton seeds is adorned with numerous vascular bundles, which are responsible for transporting assimilates produced by photosynthesis to the ovules, while the inner epidermis of the ovule is nourished by filial tissues [18,19,20]. The formation of the cotton embryo commences with the zygote’s emergence during double fertilization, followed by the central cell’s development into endosperm tissue. The key events in cotton embryogenesis unfold sequentially across three overlapping phases: morphogenesis, maturation, and desiccation [21]. The morphogenesis phase initiates with the formation of the fertilized egg and persists until 25 DPA, at which point the embryo reaches its full length [22]. Embryo development begins with the asymmetric division of the fertilized egg, yielding a small apical cell and a large basal cell, both exhibiting polarities. The apical cell evolves into the embryo, whereas the basal cell forms the suspensor, which serves as a conduit for nutrient supply to the developing embryo [23,24]. Embryo development progresses through globular, heart, and torpedo stages, with its size predominantly contingent upon the actual size of the egg cell. The maturation phase spans from 20 to 45 DPA, characterized by the rapid accumulation of oils and proteins in the cotyledons. The final phase, embryonic desiccation, commences around 35 DPA, featuring the embryo’s preparation for entering a dry, quiescent state [25]. Cotton seed embryos require substantial carbohydrates to fuel their development. Upon maturation, the dry weight of cotton fibers constitutes 40% to 50% of the total seed dry weight (inclusive of seeds and fibers), indicating a competitive relationship between cotton seed embryos and fibers for assimilates transported from photosynthesis [26]. This competitive dynamic significantly impacts the development and ultimate yield of cotton seeds.

To date, genetic research pertaining to the regulation of cotton seed size has been relatively sparse, with only a handful of genes such as *DA1* [27], *GW2* [28], *GRDP1* [29], and *SAP* [30] identified as possessing the potential to influence the size of cotton seeds. To further elucidate the molecular regulatory mechanisms governing seed size in cotton and to explore the critical genes affecting cotton seed yield and quality, this study selected two cotton materials with significant phenotypic differences, namely, N10 and N12. Through phenotypic analysis under multiple environmental conditions, these two materials exhibited notable differences in seed size. In this study, we compared the transcriptome data for N10 and N12 to unearth potential candidate genes affecting seed size. Through in-depth analysis, we identified a candidate gene named *GhUXS5*, which showed significant differential expression between the two materials, suggesting that it may play a role in regulating seed size in cotton. The findings of this study provide valuable insights for the identification of candidate genes and an understanding of the molecular mechanisms that regulate cotton seed size.

## 2. Results

### 2.1. Obtaining and Phenotypic Analysis of Cotton Seed Size Materials

Based on the data pertaining to the size traits of cotton seeds collected from four locations (Alaer and Shihezi in Xinjiang, Anyang in Henan, and Xingtai in Hebei) during 2017–2018, two lines, designated as N10 and N12, were identified. These materials exhibited significant differences in several cotton seed traits, including the weight of a hundred seeds (seed index), seed area, seed perimeter, seed length, seed width, and seed diameter (Table 1 and Table 2). Given the presence of a hull around the cotton seeds, additional measurements were taken for the traits of the cotton kernels to further verify the seed characteristics. A correlation analysis showed an extremely high degree of correlation among the various size traits of the seeds, indicating that the presence of a hull does not interfere with the study of cotton seed size traits (Appendix A).

Following multiple generations of self-pollination, the two cotton lines, N10 and N12, maintained their significant phenotypic differences in seed size (Figure 1a–d). Specific data revealed that the hundred-seed weight (seed index) of N10 was only 58.8% of that of N12. In terms of morphological parameters such as seed area, perimeter, length, width, and diameter, the values for N10 were 63.4%, 79%, 79.2%, 81.7%, and 79.5% of those for N12, respectively (Figure 1e–j). These data clearly illustrate the significant differences in seed size phenotypes between N10 and N12, providing ideal genetic materials for studying the genetic regulation of cotton seed size.

### 2.2. Transcriptome Assembly and Sample Clustering

To understand the changes at the transcription level, RNA-seq of seeds in different developmental stages (5, 20, 30, and 35 DPA) was carried out. A total of 496.37 million reads were generated in this study (Appendix A). From each sample, 148.48 Gb was obtained on average. The percentage for Q20 and Q30 was above 95% and 86%, respectively. These findings demonstrated that the quality of RNA-Seq was suitable for further investigation.

The results of principal component analysis (PCA) indicated that the two lines merged together at 5 DPA (Figure 2a). However, at 20 DPA, the two lines exhibited a more distant clustering relationship, suggesting that 20 DPA may be a critical period influencing seed size traits.

### 2.3. Cottonseed Growth Curve and Gene Expression Network

We conducted in vitro culture experiments on ovules harvested from N10 and N12 on the day of flowering. The lengths of cotton seeds at various developmental stages were measured, and growth curves were constructed (Figure 2b). The results revealed a significant divergence in growth rates between N10 and N12 at 20 DPA. Subsequent PCA pinpointed 20 DPA as a critical juncture where the seed size differences between the two genotypes become pronounced. This discovery offers vital insights into the growth dynamics and genetic regulation of cotton seed size development.

We then further compared a gene expression network specific to cotton seed size by amalgamating RNA-seq data with recognized pathways known to modulate seed size in *Arabidopsis* (Figure 2c). In the context of the IKU signaling pathway, the expression of *ABI5*, *SHB1*, and *IKU2* was robust during the initial stage, particularly at the 20 DPA phase, where the expression magnitude in N10 was notably superior to that in N12. The expression of *IKU1* and *MINI3* escalated at the later stages, particularly noticeable at 30 DPA. In the ubiquitin-proteasome signaling pathway, *DA1*, *DA2*, *EOD1/BB*, and *SOD2/UBP15* exhibited high expression at the 30–35 DPA phase, with a more pronounced expression level in N12. Similarly, within the G-protein signaling pathway, the expression levels of *AGG3*, *GPA1*, and *AGB1* in N12 were also higher than in N10 at the 30–35 DPA phase. Within the transcriptional regulators’ pathway, *DPA4/NGAL3*, *KLU*, *EOD3*, *TTG2*, and *AP2* manifested higher expression at the incipient stage of ovule development, where KLU showcased higher expression in N12, while *EOD3* and *AP2* exhibited higher expression in N10. *SOD7/NGAL3* featured a more significant expression level at the 30–35 DPA stage; its expression level in N12 was markedly superior to that in N10. In the context of the hormone signaling pathway, *ANT* and *ARF2*, relevant to auxin, demonstrated higher expression levels at the 20–30 DPA phase, while AHKs associated with cytokinins manifested higher expression in the initial stage, with their expression levels in N10 surpassing those in N12.

### 2.4. Transcriptome Differences between N10 and N12 during Seed Development

The variations in gene expression were examined through the comparison of the four different seed developmental stages, using thresholds of more than log2 (fold change) ≥ 2 and adjusted *p*-value less than 0.05. The highest number of differentially expressed genes (DEGs) was observed at the 20 DPA stage, totaling 15,646 (Appendix A). This was followed by 35 DPA (5567 DEGs) and 30 DPA (5561 DEGs). The lowest number of DEGs was noted at 5 DPA, with merely 375 DEGs. At the 20 DPA and 30 DPA stages, the number of upregulated genes surpassed that of downregulated genes, whereas the opposite was observed at the 5 DPA and 35 DPA stages (Appendix A).

### 2.5. Temporal Pattern Clusters of N10 and N12

We clustered the gene expression profiles for all developmental stages. A total of nine distinct clusters of temporal patterns were observed (Figure 3), each indicative of different expression kinetics and suggesting unique regulatory mechanisms. Notably, Clusters 4, 5, and 6 displayed relatively consistent expression patterns in both the N10 and N12 lines, regardless of time points. This indicates that these genes have relatively consistent functions and play similar roles in both lines. In contrast, Clusters 3, 7, and 8 exhibited higher expression levels in N12 compared to N10 at the 20 DPA stage, while the opposite trend was observed for Clusters 2 and 9. Furthermore, Cluster 1 demonstrated high expression in N12 at 35 DPA. This suggests that these genes may play distinct roles in the formation of cottonseed morphology.

### 2.6. Establishment of Weighed Gene Co-Expression Network Analysis (WGCNA)

Seed maturation is an intricate process of biological activity, orchestrated by a vast array of functional gene networks. These networks may coalesce into a variety of gene regulatory modules (GRMs), which collaborate in a concerted effort throughout the entire maturation sequence. The WGCNA [31] unveiled a spectrum of 15 GRMs (Figure 4a,b), each mirroring the intricate tapestry of biological processes inherent in seed maturation. Within this array of modules, the turquoise module stood out as the most expansive, boasting a repertoire of 8618 genes, in stark contrast to the midnight-blue module, which was identified as the most diminutive, comprising a mere 37 genes. These insights not only cast light on the heterogeneity of gene regulation intricacies during seed maturation but also furnish an invaluable compendium for further exploration into the genetic orchestration underlying seed development.

To elucidate the roles of various gene regulatory modules (GRMs) in seed development, we embarked on an analysis to discern the correlation between GRMs and phenotypic expressions at four distinct temporal junctures, focusing on large- and small-seed phenotypes. Our investigation revealed a multifaceted pattern of association between GRMs and phenotypic manifestations. Notably, the turquoise GRM demonstrated a pronounced correlation with the phenotype at 5 DPA, whereas the magenta, yellow, green-yellow, and midnight-blue GRMs were predominantly linked with the phenotype at 30 DPA. The salmon, purple, and blue GRMs were found to be associated with the phenotype at 35 DPA. The black and brown GRMs exhibited a significant correlation with the large-seed phenotype N10 at 20 DPA, while the cyan, green, tan, and pink GRMs were distinctly correlated with the small-seed phenotype N12 at the same developmental stage. The red GRM was observed to correlate with both phenotypes at 20 DPA. This intricate web of correlations underscores the complexity inherent in the processes of seed formation, development, and maturation, implicating a diverse array of biological pathways including substance accumulation and hormone regulation. This complexity gives rise to the emergence of distinct modules at varying developmental milestones, highlighting the intricate interplay between genetic regulation and phenotypic expression in seed development.

### 2.7. Cottonseed Size Candidate Gene Screening

To gain insight into the genetic determinants of cotton seed size traits, our study concentrated on the pivotal 20 DPA stage, marked by the most pronounced growth rate disparities and differential gene expressions between the N10 and N12 lines. Through meticulous gene expression profiling, we observed that Cluster 9 displayed the most significant expression trend divergences between the two lines at this crucial stage. Further, employing WGCNA, we pinpointed a black GRM tightly linked with N10 at 20 DPA and significantly associated with variation in cotton seed size, and we thus deemed it the most pivotal GRM influencing this trait. Subsequent to these analyses, we identified 413 candidate genes for further scrutiny.

To explore the roles and regulatory mechanisms of these candidate genes during fiber development, we conducted comprehensive gene ontology (GO) and Kyoto Encyclopedia of Genes and Genomes (KEGG) analyses. The GO enrichment analysis sorted the candidate genes into three main categories: biological processes, cellular components, and molecular functions (Figure 4c). Within the biological processes, these genes predominantly enriched categories related to cellular processes, metabolism, and hormonal activity. Concerning molecular functions, the genes showed a significant enrichment in terms of catalytic activity and binding, highlighting substantial genetic variations in these processes between the two materials. Our KEGG enrichment analysis indicated significant enrichment of these genes across 18 pathways (Figure 4d), predominantly within metabolic pathways, suggesting that the principal differences between N10 and N12 may reside in metabolic activities. Among these, a notable cluster of genes was enriched in global and overview maps, lipid metabolism, carbohydrate metabolism, and amino acid metabolism. Specifically, within the carbohydrate metabolism pathway, crucial to seed size development, 18 genes were enriched.

Among these, we focused on *Gh_D03G144400*, annotated as *GhUXS5*, which encodes UDP-glucuronate decarboxylase 5 (Figure 4e). This enzyme plays a vital role in the biosynthesis of the core tetrasaccharide integral to glycosaminoglycan biosynthesis. Notably, *GhUXS5* was markedly upregulated in N10 at 20 DPA, with diminished expression at other stages, positing it as a key candidate gene potentially pivotal in influencing cotton seed size.

### 2.8. GhUXS5 Is a Candidate Gene That Affects Seed Size Function

To probe the functional role of *GhUXS5* in seed development, we employed the GV3101 strain of *Agrobacterium tumefaciens* to generate overexpression (OE) lines of *GhUXS5* in *Arabidopsis thaliana* via the floral dip method. Subsequently, we selected three distinct *Arabidopsis GhUXS5*-OE lines for phenotypic assessment (Figure 5a). Comparative analysis between the *GhUXS5*-OE and the wild type (WT) revealed that the seeds from the OE lines exhibited a significant increase in weight (Figure 5b). Additionally, morphological differences were observed in the seeds of the OE lines compared to the wild type, with notable increases in length, width, and overall area (Figure 5d–f). Collectively, these findings suggest that the expression of *GhUXS5* may enhance seed development, indicating its potential as a genetic resource for seed size improvement.

## 3. Discussion

Cotton holds an indispensable position in global agricultural production; it is not only foundational to the textile industry but also an essential component of the edible oil sector. The fibers produced from the outer growth of the cotton ovule provide the raw materials for textiles, while the oil and protein contained within the seeds provide key raw materials for the food and feed industries. Accordingly, both cotton fiber and seed are pivotal to societal advancement. However, compared to major field crops such as wheat, rice, and maize, where the regulatory pathways governing seed size are well researched and extensively documented, studies on the regulatory pathways of cotton seeds are relatively limited and remain far from comprehensive. In this investigation, we selected two cotton materials with notable differences in seed size, N10 and N12, to probe deeply into the regulatory mechanisms of seed size. Given the complex structure of cotton seeds, comprising both the hull and the kernel, our study meticulously measured both parts separately, including parameters such as the hundred-seed weight, grain length, grain width, area, perimeter, and diameter. Correlation analyses affirmed a high degree of association between the cotton seed and the kernel, suggesting that the influence of the cotton hull on seed size is relatively minor. This discovery implies that the impact of the cotton hull can be reliably discounted when analyzing seed size. Subsequent transcriptomic analyses were conducted at different developmental stages (ovules at 5, 20, 30, and 35 DPA), and ultimately the *GhUXS5* gene was identified as a significant candidate gene affecting seed size.

The embryogenesis of cotton can be delineated into three overlapping stages: the morphogenesis period (0–25 DPA), the maturation period (20–45 DPA), and the desiccation period (post-35 DPA). To investigate the variations in gene expression throughout these stages, and particularly their influence on the regulation of seed size, this study pinpointed several critical junctures for transcriptomic analysis, including 5 DPA marking morphogenesis, 20 DPA initiating maturation, 30 DPA transitioning to desiccation, and 35 DPA commencing the dry-down phase. This intermittent sampling strategy was devised to focus on analyzing the patterns of gene expression at each pivotal phase of cotton embryo development. Employing principal component analysis (PCA) and differential expression analysis, we discerned significant disparities in the gene expression profiles of materials N10 and N12 at 20 DPA. Subsequent ex vitro culture experiments confirmed that 20 DPA represents a critical period for notable differences in seed size between the two materials. This discovery indicates that the variance in gene expression at the onset of the maturation period may have a decisive impact on the results for cotton seed size.

Given the current limited understanding of the pathways governing cotton seed development, this study endeavored to construct a gene expression network for cotton seed development by drawing upon the expression networks of seed size development in *Arabidopsis thaliana*. However, the findings revealed that despite our attempts to infer the regulatory network of cotton seed development through homologous genes, we were unable to directly identify key genes affecting seed size. Moreover, the majority of gene expression trends in the cotton seed transcriptome were the opposite of those observed in *Arabidopsis*, a phenomenon that may reflect the genetic complexity of cotton as a tetraploid plant. To overcome this challenge, we employed differential expression analysis, clusters of temporal patterns, and weighted gene co-expression network analysis (WGCNA) to screen 413 candidate genes that potentially influence cotton seed size from the transcriptome data. Through further Gene Ontology (GO) analysis and Kyoto Encyclopedia of Genes and Genomes (KEGG) pathway analysis, we identified the *GhUXS5* gene located in the carbohydrate metabolism pathway as a key candidate gene affecting cotton seed size. The discovery of *GhUXS5* provides a new perspective for understanding the molecular mechanisms of seed size regulation and may offer a new target for cotton breeding. Given the crucial role of carbohydrate metabolism pathways in plant growth and development, the function of *GhUXS5* in this pathway may have significant implications for development and maturation processes in cotton seeds.

In dicot plants, xylose serves as an important component of cell wall hemicellulose and pectin polysaccharides [32]. The biosynthesis of UDP-xylose occurs through a two-step reaction process. Firstly, UDP-glucose is converted to UDP-glucuronic acid (UDP-GlcA) in the presence of UDP-glucose dehydrogenase (UGD). Subsequently, the enzyme UDP-glucuronic acid decarboxylase (UXS) catalyzes the irreversible decarboxylation of UDP-GlcA to generate UDP-xylose [33]. UXS plays a crucial role in the interconversion of nucleotide sugar, and its activity has been observed in higher plants. The cloning of the *UXS* gene was first accomplished in the fungus *Cryptococcus neoformans* [34]. In tobacco, multiple isoforms of UXS have been isolated, and their expression has been identified in tissues associated with secondary cell wall development. Moreover, the downregulation of certain genes through antisense expression is associated with reduced levels of xylan in these tissues [35]. In *Arabidopsis*, six isoforms of UXS protein have been isolated, of which three, namely, UXS3, UXS5, and UXS6, have been found to encode enzymes localized in the cytoplasm. A significant reduction in secondary cell wall thickening is observed when these UXS isoforms are downregulated or mutated [36,37]. Additionally, UXS3 has been demonstrated to affect the accumulation of indole-3-acetic acid (IAA) in rice [38]. Therefore, *GhUXS5* may influence cotton seed size by affecting cell wall changes and the content of IAA. Furthermore, our laboratory has discovered through other studies that this gene also impacts the elongation of cotton fibers [39], indicating its multifunctionality and broad applicability.

## 4. Materials and Methods

### 4.1. Plant Materials and Sampling

Two distinct cotton lines, N10 (characterized by large seeds) and N12 (distinguished by small seeds), were sourced from a reconstructed population of recombinant inbred lines (RILs). These lines were subjected to cultivation across eight diverse environments, encompassing locations such as Alaer and Shihezi in Xinjiang, Anyang in Henan, and Xingtai in Hebei, during the years 2017 and 2018, with an additional test conducted in Anyang, Henan, in 2023. Upon the maturation of the cotton bolls, manual harvesting was performed, followed by the extraction of seeds with lint through the processes of ginning and delinting utilizing concentrated sulfuric acid. From each line, a selection of one hundred seeds was meticulously delinted. The Wanshen SC-G automatic seed tester was then employed to accurately assess six attributes of seed and kernel size for both the N10 and N12 lines under varying environmental conditions and capture photographs. The primary seed traits evaluated included the hundred-seed weight (HSW, also called seed index in cotton, g), seed length (SL, mm), seed width (SW, mm), seed area (SA, mm^2^), seed perimeter (SG, mm), and seed diameter (SD, mm). The kernel traits primarily encompassed the hundred-kernel weight (HKW, g), kernel length (KL, mm), kernel width (KW, mm), kernel area (KA, mm^2^), kernel perimeter (KG, mm), and kernel diameter (KD, mm). To conduct a comparative analysis of the cotton seed size traits between N10 and N12, a Student’s t-test was implemented.

### 4.2. RNA Extraction, Library Construction, and RNA-Seq Analysis

In 2023, the two lines were grown in field plots using a paired experiment with three replications on an experimental farm, Anyang, Henan, in 2023. RNA was extracted from ovules at 5, 20, 30, and 35 DPA and tagged for RNA sequencing (RNA-seq). The ovules were meticulously transported into a grinding vessel and swiftly submerged in liquid nitrogen. This step guaranteed consistent low-temperature processing through the consistent addition of more liquid nitrogen.

Subsequent to the collection, ovules were carefully excised using a pestle and tweezers, ensuring the integrity of each specimen. Each ovule fragment was then isolated and thoroughly triturated into a fine powder using a pestle. Following this, approximately 100 mg of the powdered sample was carefully apportioned into pre-chilled 2.0 mL centrifuge tubes, which were then promptly preserved in liquid nitrogen, awaiting subsequent analysis.

Subsequently, total RNA was isolated utilizing the FastPure Plant Total RNA Isolation kit (provided by Nanjing Vazyme Biotech Co., located in Nanjing, China), specifically designed for samples abundant in polysaccharides and polyphenolics, provided by Nanjing. The isolation procedure meticulously followed the guidelines provided by the kit’s producer. The integrity of RNA was evaluated through visual examination of the fractionated 18S and 28S rRNA bands, which were separated via electrophoresis on a 1.5% agarose gel and applied to the isolated total RNAs. Subsequently, the concentration of RNA was quantified using the Nanodrop 2000 spectrophotometric device (produced by Thermo Fisher Scientific, located in Waltham, MA, USA) by assessing the optical density ratio at 260 nm and 280 nm (OD260/OD280). Altogether, eight libraries of cDNA (comprising two distinct genotypes across four stages of development) were assembled and subsequently sequenced on the Illumina NovaSeq 6000 platform (provided by Illumina, located in San Diego, CA, USA) to analyze the transcriptomic profile.

After transcriptome sequencing, the raw data were filtered by BMKCloud (BMK, Qingdao, China). During the process of filtering the sequences, any sequences containing adapters, reads that possessed over 5% bases denoted as ‘N’, and reads where more than 20% was of low quality (quality score ≤ 15) were excluded from the subsequent analysis. The filtered sequences were aligned to the *G. hirsutum* TM-1 reference genome [40], using the Hisat2 alignment tool (v1.34d) with standard parameters [41]. The aligned files were treated by samtools (v1.15.1), and the transcription abundance of the whole genes was quantitatively estimated by StringTie (v2.2.3) [42]. We have submitted all the sequenced raw datasets to the NCBI short read archives (SRA; accession number, PRJNA1120209).

### 4.3. Principal Component Analysis, Ovule In Vitro Culture, and Growth Curves

A principal component analysis (PCA) was implemented on the transcriptome data compiled from two samples spanning four distinct stages, utilizing the OmicShare tool (https://www.omicshare.com/tools, accessed on 24 May 2024). In the proceeding steps, ovules harvested at 0 DPA were subjected to disinfection via a 0.1% mercury (II) chloride solution for a duration of 5 min. The boll shell was then carefully dissected using a surgical knife, followed by the meticulous extraction of the ovules using sterilized tweezers and their placement in a detached culture medium. Subsequently, these ovules were incubated in the absence of light at a temperature of 30 °C, enabling the observation of their growth over time. In addition, the dimensions of the cotton seeds were accurately measured, and the data were graphically represented using R (4.1.2), providing a visually intuitive depiction of the developmental trajectory.

### 4.4. Gene Expression Network

Drawing from Li’s (2019) [17] research on the molecular network that regulates plant seed size, we utilized Blast to screen for the cotton genes that showed the highest homology with the genes controlling seed size in *Arabidopsis*. Subsequently, based on the transcriptome data, we selected the cotton genes exhibiting the highest expression levels. Ultimately, we constructed a gene expression network comprising six pathways that regulate seed size, incorporating a total of 27 genes.

### 4.5. Differential Expression Analysis

In order to analyze differentially expressed genes (DEGs), we employed DESeq2 (v1.20.0)and calculated Padj values to adjust the threshold of *p* values. We defined genes with |log2FoldChange| ≥ 2 and Padj ≤ 0.01 as differentially expressed genes. Based on the transcriptional abundance of genes between the two varieties, we defined genes with higher expression levels in the large-seed variety during ovule development as upregulated genes, and those with lower expression levels in the large-seed variety as downregulated genes.

### 4.6. Clusters of Temporal Patterns

Genes exhibiting an FPKM value exceeding 5 at a particular time point were classified as expressed genes. Utilizing this criterion, a total of 38,947 expressed genes were identified. The fuzzy c-means algorithm [43] was employed to cluster gene expression profiles across all developmental stages. Drawing upon the transcriptional abundance of the two materials under investigation, an analysis was performed using the TCseq package within the R (4.1.1). The culmination of this analysis yielded nine distinct clusters, each representing a unique transcriptomic model.

### 4.7. WGCNA Pipeline

Gene co-expression networks were established through the execution of Weighted Gene Correlation Network Analysis (WGCNA) using the pertinent R (4.1.1) software package [31]. Genes with a maximum value among eight samples smaller than five were excluded. The remaining genes were selected for the construction of the weighted gene co-expression network. A soft-thresholding criterion was chosen based on an R2 value surpassing 0.85. The parameter for merging modules, referred to as merge Cut Height, was set at a level of 0.35 to facilitate module classification. To evaluate the association between modules and phenotypes, the Pearson correlation coefficient between the module’s eigengene and the phenotype was calculated using the statistics package in Python (v3.9.0). For the construction of gene regulatory networks (GRNs), Pearson correlations between the expression levels of the target gene and other candidate genes were computed. The GRNs were documented in the form of tab-separated tables. In this study, functional enrichment analysis was conducted using GO and KEGG enrichment analysis via cotton FGD (https://cottonfgd.net/, accessed on 3 May 2024).

### 4.8. Phenotypic Identification of Arabidopsis Overexpressed GhUXS5

The primer pair *GhUXS5*-F and *GhUXS5*-R (Table 3) were used to amplify the 1041 bp coding sequence (CDS) of *GhUXS5*. Subsequently, using Gateway cloning technology and the BP and LR Clonase enzyme mixtures (Thermo Fisher Scientific, Zhengzhou, China), the CDS of *GhUXS5* was integrated into the pEarleyGate 101 plasmid to generate an overexpression vector. The prepared vector was then introduced into the *Agrobacterium tumefaciens* strain GV3101 (Weidi Biotechnology, Shanghai, China), and the recombinant GV3101 plasmid was used to infect *Arabidopsis thaliana* (Col-0).

Seeds reaped from the T3 generation of *Arabidopsis*, comprising both overexpressing specimens and those of wild type, were utilized for the quantification of the thousand-seed weight, with each instance encompassing the enumeration of 1000 seeds, executed thrice for repetition. Subsequently, the dimensions of the seeds were captured photographically via a stereomicroscope. Ten seeds from an identical field of view were selected, and their respective lengths, widths, and surface areas were measured utilizing ImageJ (v1.8.0.345) software.

## 5. Conclusions

In this study, the transcriptomic analysis of two lines, N10 and N12, during cotton embryo development was conducted to systematically screen 413 genes as candidate genes influencing seed size, ultimately identifying *GhUXS5* as a key candidate gene. Through GO and KEGG analysis, the functional role of *GhUXS5* in the carbohydrate metabolism pathway was further revealed; this gene plays a crucial role in regulating cotton seed size. Additionally, preliminary gene functional validation was performed to further confirm the importance of *GhUXS5* in cotton embryo development. The results of this study hold significant scientific value in deepening our understanding of the molecular mechanisms underlying cotton seed growth and development. By uncovering key genes and associated pathways, we can gain a better understanding of the regulatory mechanisms behind cotton seed size. This is of great significance for breeders, as they can utilize these findings to design and manipulate the desired seed size traits, thereby enhancing the economic value and adaptability of cotton crops.

## Figures and Tables

**Figure 1 ijms-25-09812-f001:**
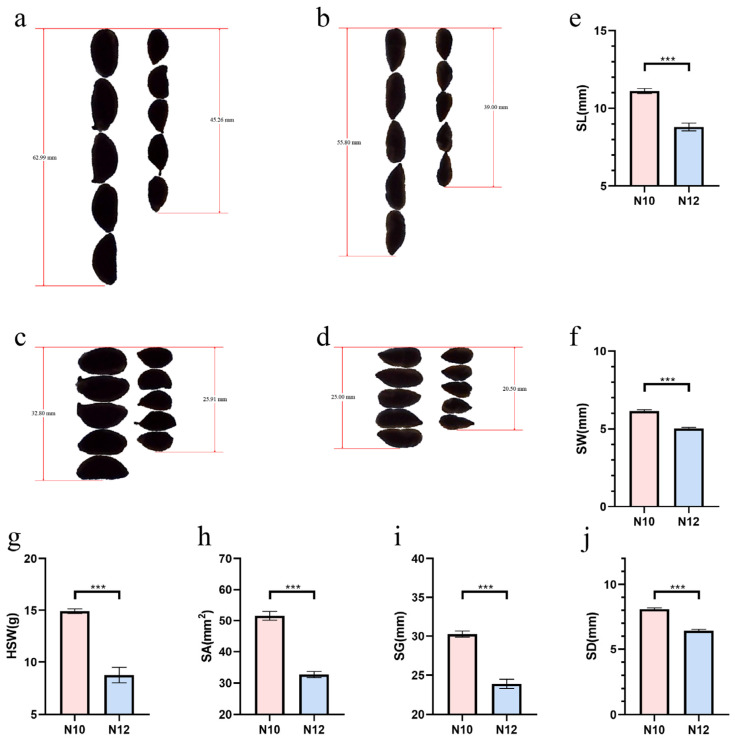
Comparison of mature cottonseed and kernels in cotton materials N10 and N12. (**a**,**b**) Comparison of mature cottonseed and kernel length in N10 and N12; (**c**,**d**) comparison of mature cottonseed and kernel width in N10 and N12; and (**e**–**j**) comparison of mature cottonseed length, width, 100-seed weight, area, perimeter, and diameter in N10 and N12. Three biological replicates were used for statistical analysis (*t*-test; *** *p* < 0.001). Values in (**e**–**j**) represent means ± SE (*n* = 3).

**Figure 2 ijms-25-09812-f002:**
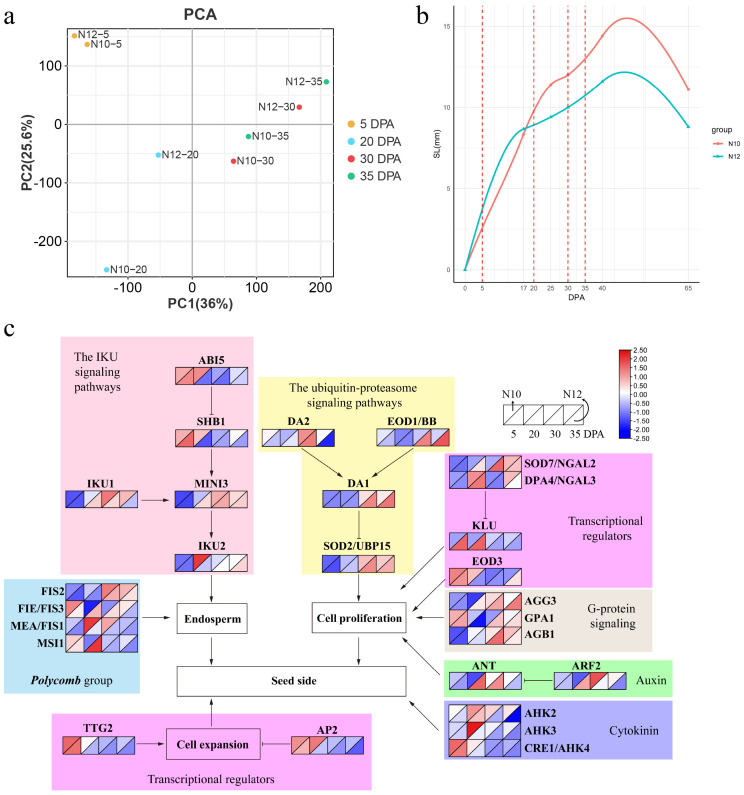
Cottonseed development analysis. (**a**) Principal component analysis (PCA) of genes identified from eight samples, with three biological replicates per sample. (**b**) Growth curve of cottonseed length. (**c**) Gene expression network for cottonseed size. The top left and bottom right represent N10 and N12 genotypes.

**Figure 3 ijms-25-09812-f003:**
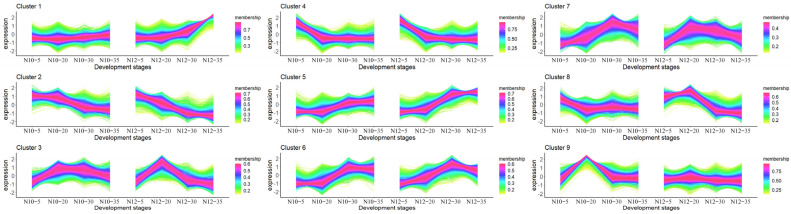
Nine different time patterns of transcriptome expression. The X-axis represents the four developmental stages (5, 20, 30, and 35 DPA) for the two materials.

**Figure 4 ijms-25-09812-f004:**
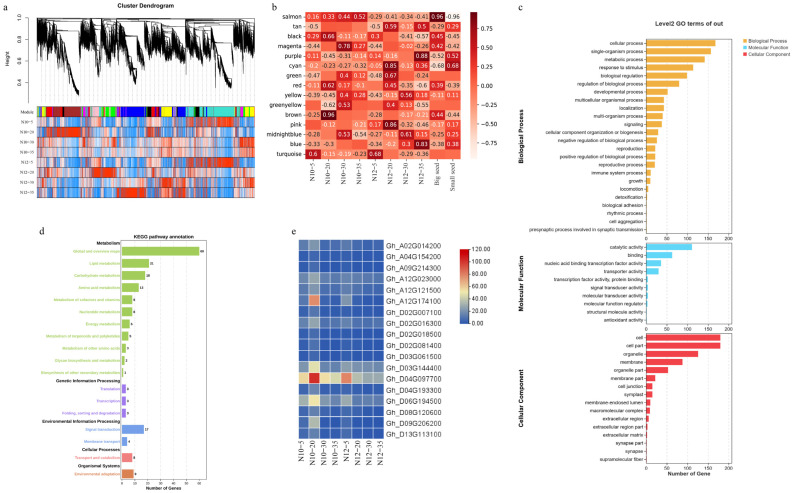
WGCNA of transcriptome data. (**a**) Module classification based on WGCNA results: the dendrogram shows the classification of gene clusters based on expression. (**b**) Heatmap of module–trait relationships. (**c**) GO enrichment analysis of the gene regulatory module (GRM) containing downregulated genes. (**d**) KEGG enrichment analysis of the GRM containing downregulated genes. (**e**) Expression patterns of candidate genes.

**Figure 5 ijms-25-09812-f005:**
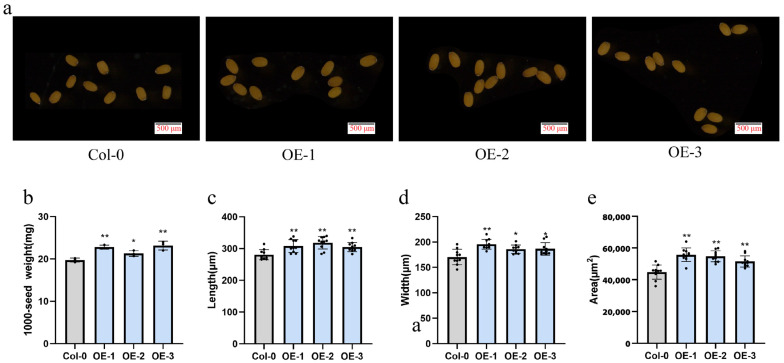
Phenotypic analysis of *GhUXS5* overexpression lines in *Arabidopsis thaliana*. (**a**) Morphology of seeds from *Arabidopsis* wild type and overexpression lines. (**b**–**e**) Thousand-seed weight, length, width, and area of seeds from *Arabidopsis* wild type and overexpression lines. Three biological replicates were used for statistical analysis (*t*-test; * *p* < 0.05, ** *p* < 0.01). Values in (**b**–**e**) represent means ± SE.

**Table 1 ijms-25-09812-t001:** Phenotypic data for cotton seed materials N10 and N12 in eight different environments.

Year	Location	HSW		SA		SG		SL		SW		SD	
		N10	N12	N10	N12	N10	N12	N10	N12	N10	N12	N10	N12
2017	Alaer	12.71	7.37	44.21	28.17	27.91	21.99	10.32	8.12	5.75	4.69	7.49	5.98
	Shihezi	12.51	7.42	45.91	28.71	28.31	22.15	10.59	8.32	5.75	4.63	7.63	6.04
	Anyang	12.82	7.48	44.89	29.41	28.19	22.61	10.51	8.39	5.66	4.78	7.55	6.10
	Xingtai	14.28	8.16	50.40	30.38	29.98	23.15	11.06	8.62	6.06	4.82	8.00	6.21
2018	Alaer	11.38	8.05	41.95	29.92	27.09	22.33	10.32	8.53	5.40	4.74	7.26	6.17
	Shihezi	10.68	6.91	43.00	25.67	27.12	20.63	10.33	7.82	5.57	4.39	7.39	5.71
	Anyang	11.41	8.21	43.80	30.69	27.66	22.77	10.39	8.58	5.61	4.86	7.46	6.24
	Xingtai	13.83	7.63	49.90	28.94	29.59	22.23	11.04	8.33	5.97	4.70	7.96	6.06

**Table 2 ijms-25-09812-t002:** Phenotypic data for kernels for cotton materials N10 and N12 in eight different environments.

Year	Location	HKW		KA		KG		KL		KW		KD	
		N10	N12	N10	N12	N10	N12	N10	N12	N10	N12	N10	N12
2017	Alaer	7.50	4.42	28.38	17.86	22.81	17.55	8.80	6.78	4.21	3.49	6.00	4.76
	Shihezi	7.41	4.64	28.59	18.92	23.22	18.64	9.01	7.18	4.19	3.50	6.02	4.90
	Anyang	8.28	5.28	30.34	20.39	23.85	18.72	9.24	7.23	4.34	3.78	6.21	5.09
	Xingtai	9.02	5.28	32.00	19.96	24.59	18.78	9.50	7.17	4.46	3.74	6.38	5.04
2018	Alaer	7.33	5.02	27.04	19.18	22.12	18.13	8.81	7.16	4.02	3.54	5.86	4.94
	Shihezi	6.17	4.33	26.10	17.52	22.72	17.45	9.00	6.73	3.86	3.46	5.76	4.72
	Anyang	6.99	5.29	26.39	20.22	22.33	18.67	8.97	7.30	3.85	3.71	5.79	5.07
	Xingtai	9.92	5.00	34.37	19.71	24.82	18.61	9.56	7.19	4.76	3.69	6.61	5.00

**Table 3 ijms-25-09812-t003:** All the primer sequences used in this research.

Primer Name	Sequence
*GhUXS5*-F	GGGGACAAGTTTGTACAAAAAAGCAGGCTTAATGGCAGCAAATTCATCAAATG
*GhUXS5*-R	GGGGACCACTTTGTACAAGAAAGCTGGGTAGTTCTCTTTGGAGACTCCGAG

## Data Availability

Data will be made available on request. All data and materials supporting our findings are included in Section 4. Details are provided in the attached files. All the transcriptome raw data that we sequenced were deposited in the NCBI short read archives (SRA; accession number, PRJNA1120209).

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
