# Peer review of "Transcriptome Analysis and Identification of Genes Associated with Cotton Seed Size"

_ijms, 2024, doi:10.3390/ijms25189812_

Round 1

Reviewer 1 Report

Comments and Suggestions for Authors

Dear authors,

In the present study, the authors have employed transcriptome analysis (RNA-seq) to identify genes associated with cottonseed size. Based on RNA-seq, 413 candidate genes, were significantly enriched in transcriptional regulation. Among them, authors identified GhUXS5 as a key candidate gene influencing seed size. To confirm that, the following, authors conducted further a functional study of GhUXS5 by overexpression of this gene in Arabidopsis, revealing its crucial role in regulating cottonseed size.

Comments and Suggestions for Authors

1. Firstly, the figure (such as Fig 2-4) data was presented at low resolution. Especially, Fig4 in there has several sub-figures, even though I increased the magnification of the entire sub-figures, but they could still not be seen clearly; hence, I could not access all the entire figures’ information. Authors should provide and replace them with high-resolution ones.

2. The obtained data is still poor to support for identification of genes associated with cottonseed size. The data of Table 1, Table 2, and Fig1 is not supported much in the study topic since they are phenotypic data of materials N10 (large seeds) and N12 (small seeds) with notable differences in seed size growing under different locations, given general information of study’s materials. As obtained results (Fig 4c,d), transcriptome data showed that most of these related to biological and metabolism processes, including sugar metabolism were significantly enriched. Therefore, this study will be more interesting if the authors include metabolic analysis to support finding’s research.  

3. In addition, to support the findings research, the authors should evaluate the GhUXS5 expression and other related genes that control seed size in Arabidopsis overexpressed GhUXS5. It could be supported for the results of Fig 5.

4. Based on results of Fig 4e, with expression patterns of 18 candidate genes, showed that the strongest expressed is Gh_D04G097700, followed by Gh_A12G174100, these were highly expressed at 20 DAP in the N10 cotton line. However, the authors choose Gh_D03G144400 (annotated as GhUXS5) for heterologous expression in Arabidopsis. It did not make sense.

5. Please add more description for section 4.8 by providing more detail on how GhUXS5 was isolated and cloned; which vector was used; how constructed expression vector…etc. This is important information so that the reader/researcher can access this information and replicate it based on the authors' research.

Moreover, this subsection “4.8. GhUXS5 Overexpression Arabidopsis Phenotypic Identification” should be rewritten as “4.8. Phenotypic Identification of Arabidopsis Overexpressed  GhUXS5

Minor remarks:

- Repetitive sentence: “approximately 100mg of the powdered sample was carefully apportioned into pre-chilled 2.0 ml centrifuge tubes, which were then promptly preserved in liquid nitrogen, awaiting subsequent analysis.” (Line 391-394).   

Vs. “around 100 mg of the fiber powder sample was loaded into a pre-cooled 2.0 ml centrifuge tube, which was then placed in liquid nitrogen for storage and further processing” (line 395-397).

- Italicize gene name: GhUXS5, UXS and the scientific name of plant/bacteria/fungi: Arabidopsis thaliana, Agrobacterium tumefaciens, Cryptococcus neoformans.

- Delete “after flowers at 0 DPA” (line 385).

- Present the abbreviation “DPA” instead of its full form “days post-anthesis” after 1st place of its full form was mentioned.

- Rewrite the sentence: “Despite the significant contributions of cotton seeds to national food security…” (line 295) is appropriate in terms of using “national food security”, since the cotton seeds do not serve as staple foods. It is only an important as raw material for edible oil and a source of biofuel.

- Capitalize Each Word in the paper title and subsection following other journals’ style.

For example: “2.2. Transcriptome assembly and sample clustering” -> “2.2. Transcriptome Assembly and Sample Clustering”

- Please note on citation number

- Reformat the style of Reference according to the requirements of the journal

“Author 1, A.B.; Author 2, C.D. Title of the article. Abbreviated Journal Name YearVolume, page range.”

For example, Ref 2 (line 517):

Ruan, Y. L., Recent advances in understanding cotton fibre and seed development. Seed Sci. Res. 2005, 15, (4), 269-280.

-> Ruan, Y. L., Recent advances in understanding cotton fibre and seed development. Seed Sci. Res. 2005, 15, 269‒280.

I have marked the above minor remarks/comments on the manuscript. Please use it for easy tracking to revise.

Author Response

Dear Reviewer,

Thank you very much for giving us the opportunity to revise our manuscript. On behalf of all the authors, I wish to thank you for taking out your valuable time to review this manuscript and giving the valued comments and suggestions which should substantially improve this manuscript. We have addressed the individual items of your concerns and suggestions as below.

Comments 1: Firstly, the figure (such as Fig 2-4) data was presented at low resolution. Especially, Fig4 in there has several sub-figures, even though I increased the magnification of the entire sub-figures, but they could still not be seen clearly; hence, I could not access all the entire figures’ information. Authors should provide and replace them with high-resolution ones.

Response 1: Thank you for pointing this out. I uploaded a Word version on the website, and the automatically generated PDF resulted in unclear images. Next, I will upload a PDF file containing clear images.

Comments 2: The obtained data is still poor to support for identification of genes associated with cottonseed size. The data of Table 1, Table 2, and Fig1 is not supported much in the study topic since they are phenotypic data of lines N10 (large seeds) and N12 (small seeds) with notable differences in seed size growing under different locations, given general information of study’s lines. As obtained results (Fig 4c,d), transcriptome data showed that most of these related to biological and metabolism processes, including sugar metabolism were significantly enriched. Therefore, this study will be more interesting if the authors include metabolic analysis to support finding’s research.

Response 2: Thank you for the comment. In this study, we used multi-environment data to identify two cotton lines with significant differences in seed size. In each environment, we measured more than 100 seeds to support this result. We believe that there are genes affecting seed size in these two significantly different lines, and we aim to identify key genes through transcriptome analysis. This approach has also been used in other articles, including studies on other plants (Garg R et al.,2017 ) or other traits of cotton seeds ( Zhu et al.,2021 ). Additionally, the idea you suggested of conducting research through metabolism is very good, and I will apply and verify it in future studies.

Garg R, Singh VK, Rajkumar MS, Kumar V, Jain M. Global transcriptome and coexpression network analyses reveal cultivar-specific molecular signatures associated with seed development and seed size/weight determination in chickpea. Plant J. 2017 Sep;91(6):1088-1107. doi: 10.1111/tpj.13621. Epub 2017 Aug 1. PMID: 28640939.

Zhu D, Le Y, Zhang R, Li X, Lin Z. A global survey of the gene network and key genes for oil accumulation in cultivated tetraploid cottons. Plant Biotechnol J. 2021 Jun;19(6):1170-1182. doi: 10.1111/pbi.13538. Epub 2021 Jan 19. PMID: 33382517; PMCID: PMC8196633.

Comments 3: In addition, to support the findings research, the authors should evaluate the GhUXS5 expression and other related genes that control seed size in Arabidopsis overexpressed GhUXS5. It could be supported for the results of Fig 5.

Response 3: Thank you for the comment. In this study, the genes regulating seed size in Arabidopsis thaliana showed no significant differences in the two cotton lines. It is speculated that there are genomic differences between Arabidopsis thaliana and cotton, and the genes regulating seed size are not the same. Therefore, we used transcriptome data to identify GhUXS5 as a potential gene affecting cottonseed size. However, there are few studies on cottonseed size, and no clear regulatory pathways have been identified. We have not conducted qRT-PCR detection of other genes in transgenic Arabidopsis thaliana.

Comments 4: Based on results of Fig 4e, with expression patterns of 18 candidate genes, showed that the strongest expressed is Gh_D04G097700, followed by Gh_A12G174100, these were highly expressed at 20 DAP in the N10 cotton line. However, the authors choose Gh_D03G144400 (annotated as GhUXS5) for heterologous expression in Arabidopsis. It did not make sense.

Response 4: Thank you for the comment. We determined that 20 days post-anthesis (DPA) is a critical period using the cottonseed growth curve, and the developmental process of the two lines is basically similar in the early stages of cottonseed development. Therefore, we did not choose Gh_D04G097700, which showed differences at 5 DPA. Additionally, the gene function annotation of Gh_A12G174100 is Glutamate decarboxylase 4, and this gene is not related to seed development. GhUXS5 shows significant differences only at 20 DPA, and there are reports that UXS regulates seed development by affecting the cell wall. In conclusion, we chose GhUXS5 as the key gene regulating cottonseed size.

Comments 5: Please add more description for section 4.8 by providing more detail on how GhUXS5 was isolated and cloned; which vector was used; how constructed expression vector…etc. This is important information so that the reader/researcher can access this information and replicate it based on the authors' research.

Moreover, this subsection “4.8. GhUXS5 Overexpression Arabidopsis Phenotypic Identification” should be rewritten as “4.8. Phenotypic Identification of Arabidopsis Overexpressed GhUXS5

Response 5: I agree with your opinion and have added the method for constructing transgenic Arabidopsis thaliana plants in section 4.8 of the methods. The title of section 4.8 has also been changed.

The minor remarks section has also been modified according to your suggestions. Thank you again for your careful and meticulous review of the article.

Reviewer 2 Report

Comments and Suggestions for Authors

Please find my review for the manuscript ID: ijms-3182691 entitled as “Jia et al., Transcriptome analysis and identification of genes associated with cotton seed size.

The manuscript is well written with convincing data and figures generation. The results and discussion parts are well interpreted.

Some concerns, corrections and questions:  

1.     Figure 2c labeling and presentation with figure’s legend is difficult to understand. Please clearly label the right genotype (N10 and N12) with its corresponding figure part. For instance, the legend written as “The top left and bottom right represent N10 and N12 genotypes” line#150-151 is confusing. Why don’t you label the figure parts with its appropriate genotype (N10 and N12)?

2.     Sentence between line#143-144 in not clear. It seems that something missing. Check it!

3.      I am curious for sequence differences (coding region) of GhUXS5 gene between N10 and N12. Do you have any sequence data for any natural mutation events in N12?

4.     Why don’t support your RNA-seq DEGs of GhUXS5 in N10 and N12 by real-time quantitative PCR analysis for further scientific validation?

Thank you!

Author Response

Dear Reviewer,

Thank you very much for giving us the opportunity to revise our manuscript. On behalf of all the authors, I wish to thank you for taking out your valuable time to review this manuscript and giving the valued comments and suggestions which should substantially improve this manuscript. We have addressed the individual items of your concerns and suggestions as below.

Comments 1: Figure 2c labeling and presentation with figure’s legend is difficult to understand. Please clearly label the right genotype (N10 and N12) with its corresponding figure part. For instance, the legend written as “The top left and bottom right represent N10 and N12 genotypes” line#150-151 is confusing. Why don’t you label the figure parts with its appropriate genotype (N10 and N12)?

Response 1: Thank you for pointing this out. We have made modifications to Figure 2c and added a legend.

Comments 2: Sentence between line#143-144 in not clear. It seems that something missing. Check it!

Response 2: Thank you for the comment. We have made modifications to line#143-144, changing the original sentence "The results of principal component analysis (PCA) indicated that the two lines ed together during the same period (Figure 2a)." to " The results of principal component analysis (PCA) indicated that the two lines merged together at 5 DPA (Figure 2a). ".

Comments 3: I am curious for sequence differences (coding region) of GhUXS5 gene between N10 and N12. Do you have any sequence data for any natural mutation events in N12?

Response 3: Thank you for the comment. We cloned the cDNA sequences of the two lines and found no differences. We inferred that the differences in the expression levels of GhUXS5 might be due to variations in the promoter region. Research on the promoter will be the focus of our next study.

Comments 4: Why don’t support your RNA-seq DEGs of GhUXS5 in N10 and N12 by real-time quantitative PCR analysis for further scientific validation?

Response 4: Thank you for the comment. We are currently preparing to conduct qRT-PCR; however, it will take a considerable amount of time and cannot be completed within the specified timeframe. We will update the manuscript with the qRT-PCR results once the experiment is completed.

Thank you again for your careful and meticulous review of the article.

Reviewer 3 Report

Comments and Suggestions for Authors

The reviewed manuscript, titled “Transcriptome analysis and identification of genes associated with cotton seed size” by Bing Jia et al. is a work that is focused on extending our understanding of the relationship between seed size and agricultural output in the extremely important and valuable economic crop of cotton.  The authors provide a clear and focused scientific analysis where they systematically characterize and screen two lineages of cotton to identify differentially expressed genes as well as their relation to seed size and outcomes.  The work is quite deliberate – in the best way, in that the authors clearly and thoroughly design, perform, and analyze their work prior to moving on.  Overall, I would like to commend the authors on this submission – I think that it represents a very high-quality submission that is quite well done and written.  As this is the case, I believe that this work is worthy of publication in IJMS and only requires minor revisions.  I would recommend two things prior to publication of the final work:

1) the figure formatting – size of fonts, etc. needs to be considered and made in the size that is legible.  The way that the figures are made is appropriate, but the formatting, while appropriate for the review copy, needs work.

2) minor copy editing is required.  Spacing is not consistent, minor typos are present, etc.  Nothing egregious.  The authors do use the word subsequently quite often, however, though that is a personal choice. 

Comments on the Quality of English Language

English is fine.  Final editing needed, but the authors seem capable of accomplishing this on their own.

Author Response

Dear Reviewer,

Thank you very much for giving us the opportunity to revise our manuscript. On behalf of all the authors, I wish to thank you for taking out your valuable time to review this manuscript and giving the valued comments and suggestions which should substantially improve this manuscript. We have addressed the individual items of your concerns and suggestions as below.

Comments 1: the figure formatting – size of fonts, etc. needs to be considered and made in the size that is legible.  The way that the figures are made is appropriate, but the formatting, while appropriate for the review copy, needs work.

Response 1: Thank you for pointing this out. We have modified the size and format of the images to enhance clarity and readability.

Comments 2: minor copy editing is required. Spacing is not consistent, minor typos are present, etc. Nothing egregious. The authors do use the word subsequently quite often, however, though that is a personal choice.

Response 2: Thank you for the comment. We have corrected some spelling and grammatical errors. Additionally, we adjusted the spacing to improve the formatting and ensure a more standardized appearance of the manuscript.

Thank you again for your careful and meticulous review of the article.

Round 2

Reviewer 1 Report

Comments and Suggestions for Authors

Dear authors,

I do agree with most of your responses. However, I am still not satisfied with some of them. Again, I’d like to bring up these things (denoted as Re-comments) that need to be clarified.

Comments 1: Firstly, the figure (such as Fig 2-4) data was presented at low resolution. Especially, Fig4 in there has several sub-figures, even though I increased the magnification of the entire sub-figures, but they could still not be seen clearly; hence, I could not access all the entire figures’ information. Authors should provide and replace them with high-resolution ones.

Response 1: Thank you for pointing this out. I uploaded a Word version on the website, and the automatically generated PDF resulted in unclear images. Next, I will upload a PDF file containing clear images.

Re-comment 1: Even though authors has replaced these Figures, but I have not noticed any improvement in them. Yes, decreased Figure quality/resolution may be caused during conversion. I think authors should consider on regeneration of high-quality original images; If these originals are already formed high-resolution, I recommend the author should upload them in zip files to journal system.

In case, author own high-quality original images, change “paste option” during insertion the contents of the clipboard as an embedded file might be help. Tips: do not “paste opions” as “Figure (U)”. [In word; Tab “Home”->”Paste”/Paste Options->Paste Special-> Paste as: Files/insertion the contents of the clipboard as an embedded file.

Comments 3: In addition, to support the findings research, the authors should evaluate the GhUXS5 expression and other related genes that control seed size in Arabidopsis overexpressed GhUXS5. It could be supported for the results of Fig 5.

Response 3: Thank you for the comment. In this study, the genes regulating seed size in Arabidopsis thaliana showed no significant differences in the two cotton lines. It is speculated that there are genomic differences between Arabidopsis thaliana and cotton, and the genes regulating seed size are not the same. Therefore, we used transcriptome data to identify GhUXS5 as a potential gene affecting cottonseed size. However, there are few studies on cottonseed size, and no clear regulatory pathways have been identified. We have not conducted qRT-PCR detection of other genes in transgenic Arabidopsis thaliana.

Re-comment 3: As authors’ response “the genes regulating seed size in Arabidopsis thaliana showed no significant differences in the two cotton lines”. I do not see any data or evident for this were presented in the present study.

Moreover, authors should conduct evaluate the GhUXS5 expression and other related genes that control seed size in the transgenic Arabidopsis overexpressed GhUXS5 instead of “We have not conducted qRT-PCR detection of other genes in transgenic Arabidopsis thaliana.” Many published studies have found that there were a various genes regulate seed size in Arabidopsis has been identified (https://doi.org/10.3390/ijms241310666; https://doi.org/10.1111/nph.15642; ect. ). Similar to that as presented in Fig2c (for cotton), there do have the gene expression network for controlling seed size in Arabidopsis. Therefore, authors should investigate how the cotton GhUXS5 can increase the seed size of transgenic Arabidopis (overexpressed lines). It could be conducted via qRT-PCR to determine the key gene involved.

Minor remarks:

Authors’ Response: The minor remarks section has also been modified according to your suggestions. Thank you again for your careful and meticulous review of the article.

Re-comment: I see some of my comments as minor remarks still have not been addressed yet.

- Italicize gene name: GhUXS5 and the scientific name of plant Arabidopsis thaliana (Fig 5’ legend, elsewhere in Section “Discussion”). Again, please carefully scanning in whole manuscript to fix it.

- Since 1st place of its full form was mentioned (line 77), hence, present the abbreviation “DPA” instead of its full form “days post-anthesis” (line 194, 382)

- Capitalize Each Word in the paper title and subsection following other journals’ style.

Some are still there: Sub-section 2.1, 4.3, 4.4

- Please note on citation number style/format: (putting space between the text and cited number of Ref).

For example: “textile industry[1]” ->“textile industry [1]”; “livestock  feed[2]” -> livestock  feed [2]”

- Represent “The prepared vector was then introduced into Agrobacterium (GV3101, Weidi Biotechnology, Shanghai, China)” (line 469-470)

-> “The prepared vector was then introduced into Agrobacterium tumefaciens strain GV3101 (Weidi Biotechnology, Shanghai, China)”.

Thank you very much for your revising and response in Round 1.

Author Response

Dear Reviewer,

Thank you very much for giving us the opportunity to revise our manuscript. On behalf of all the authors, I wish to thank you for taking out your valuable time to review this manuscript and giving the valued comments and suggestions which should substantially improve this manuscript. We have addressed the individual items of your concerns and suggestions as below.

Re-comment 1: Even though authors has replaced these Figures, but I have not noticed any improvement in them. Yes, decreased Figure quality/resolution may be caused during conversion. I think authors should consider on regeneration of high-quality original images; If these originals are already formed high-resolution, I recommend the author should upload them in zip files to journal system.

Response 1: Thank you for pointing this out. I have modified the images in the PDF again to enhance their clarity. Additionally, I have uploaded the image files in a zip format to the journal system.

Re-comment 3: As authors’ response “the genes regulating seed size in Arabidopsis thaliana showed no significant differences in the two cotton lines”. I do not see any data or evident for this were presented in the present study.

Moreover, authors should conduct evaluate the GhUXS5 expression and other related genes that control seed size in the transgenic Arabidopsis overexpressed GhUXS5 instead of “We have not conducted qRT-PCR detection of other genes in transgenic Arabidopsis thaliana.” Many published studies have found that there were a various genes regulate seed size in Arabidopsis has been identified (https://doi.org/10.3390/ijms241310666; https://doi.org/10.1111/nph.15642; ect.). Similar to that as presented in Fig2c (for cotton), there do have the gene expression network for controlling seed size in Arabidopsis. Therefore, authors should investigate how the cotton GhUXS5 can increase the seed size of transgenic Arabidopis (overexpressed lines). It could be conducted via qRT-PCR to determine the key gene involved.

Response 2: Thank you so much for your feedback on the paper. I’m really sorry for my earlier mistake when I said, “the genes regulating seed size in Arabidopsis thaliana showed no significant differences in the two cotton lines.”

I agree with your suggestion that the authors should evaluate the GhUXS5 expression and other related genes that control seed size in the transgenic Arabidopsis overexpressing GhUXS5. But unfortunately, I don't have any samples or Arabidopsis plants right now, and I also need to submit my response by September 9th, so I’m really sorry about this. Thank you for your detailed review of our study.

According to your suggestions, the minor remarks section has been modified again. Thank you once again for your careful review of the article.

Round 3

Reviewer 1 Report

Comments and Suggestions for Authors

Dear authors,

Thank you very much for your hard work and revised the manuscript.